# Efficient Cross-Modality Abdominal Organ Segmentation Using nnU-Net and MIND Descriptors

Yannick Kirchhoff[1,2,3], Maximilian Rokuss[1,3], Benjamin Hamm[1,4], Ashis Ravindran[1], Constantin Ulrich[1,4,5], Klaus Maier-Hein[1,6†], and Fabian Isensee[1,7†]

[1] German Cancer Research Center (DKFZ) Heidelberg, Division of Medical Image Computing, Heidelberg, Germany
[2] HIDSS4Health - Helmholtz Information and Data Science School for Health, Karlsruhe/Heidelberg, Germany
[3] Faculty of Mathematics and Computer Science, Heidelberg University, Heidelberg, Germany
[4] Medical Faculty Heidelberg, Heidelberg University, Heidelberg, Germany
[5] National Center for Tumor Diseases (NCT), NCT Heidelberg, A partnership between DKFZ and University Medical Center Heidelberg
[6] Pattern Analysis and Learning Group, Department of Radiation Oncology, Heidelberg University Hospital, Heidelberg, Germany
[7] Helmholtz Imaging, DKFZ, Heidelberg, Germany
`yannick.kirchhoff@dkfz-heidelberg.de`

**Abstract.** Accurate segmentation of abdominal organs in magnetic resonance imaging (MRI) is essential for diagnosis and treatment planning. However, this task is challenging due to the scarcity of labeled MRI data and significant differences in appearance between MRI and computed tomography (CT) images. Task 3 of the FLARE 2024 challenge was launched to encourage the development of algorithms capable of transferring knowledge from labeled CT scans to unlabeled MRI scans for efficient abdominal organ segmentation under strict resource constraints. In this paper, we describe our contribution to this challenge by utilizing nnU-Net combined with modality-independent neighborhood descriptor (MIND) features to transfer labels from CT to MRI. Our method achieved an average Dice Similarity Coefficient (DSC) of 57.7% and an average Normalized Surface Dice (NSD) of 59.8% on the validation set, with an average running time of 20 seconds and an area under the GPU memory-time curve of 73,607 MB. These results demonstrate that our approach effectively addresses the challenges of cross-modality abdominal organ segmentation under resource constraints, highlighting the potential of modality-independent descriptors for label transfer in medical imaging.

**Keywords:** FLARE Challenge · Organ Segmentation · nnU-Net · MIND Descriptors.

---

† Shared last authorship

## 1   Introduction

Accurate organ and lesion segmentation in medical imaging is crucial for improving diagnostic accuracy, treatment planning, and monitoring the progression of diseases. In recent years, segmentation challenges in medical imaging have driven significant advancements in algorithm development, particularly in the field of abdominal cancer segmentation. However, the task of abdominal organ segmentation on pathological scans presents unique challenges due to the wide variety of cancer types, lesion sizes, and corresponding differences in the appearances of organs.

Task 3 of the FLARE 2024 challenge builds on earlier iterations of the FLARE challenge, shifting the focus to abdominal organ segmentation on MRI images. The challenge provides a dataset consisting of 2,050 CT scans and more than 4,800 MRI scans. The provided CT scans are the same as in Task 2, comprising 50 fully labeled scans and 2,000 unlabeled / pseudolabeled scans. The MRI images on the other hand are completely unlabeled and span different sequences such as T1, T2, DWI and different contrast enhanced sequences. The main difficulty in this task is the knowledge transfer between modalities.

Moreover, hard constraints on inference VRAM usage and time limit the possible network architectures, forcing careful trade-offs between model complexity, ensembling strategies, and test-time augmentations. This necessitates efficient models that can achieve high segmentation accuracy while remaining within resource limitations.

Domain adaptation is an active area of research in the field of medical imaging. Most work in this field focuses on shifts due to different centers, imaging protocols or populations, where common test-time adaptation methods [24,10,14,3] show promising performance. However, these methods are typically not applied in the context of modality transfer. In the field of multimodal deformable image registration, MIND descriptors [7,8] are used to obtain a modality-independent representation of an image.

This manuscript describes our approach for abdominal organ segmentation on MRI images, learning from CT images in Task 3 of the FLARE 2024 challenge. We employ nnU-Net [11] with modifications to achieve efficient inference and adhere to resource and time constraints during inference. MIND descriptors are used to transfer labels from the CT images to the MRI images.

## 2   Method

Our contribution builds upon the state-of-the-art nnU-Net framework [11]. Due to the time and resource constraints imposed during inference, we cannot use the proposed default U-Net configuration, let alone the newly proposed ResEncL configuration [12].

## 2.1 Proposed Method

Our method consists of multiple steps. First, we train a default nnU-Net on the MIND descriptors [7,8] of the 50 labeled CT scans and use this model for inference on the MIND descriptors of the T1 images from the unlabeled MRI dataset. In the next step, we filter the 1,331 T1 images to select images with all 13 organ labels present, which leaves us with 99 images for training of the next model. This model, trained specifically on T1 images, is then used to generate labels for the missing 1,232 T1 images. Finally, the labels are transferred to the rest of the images through affine transformation, to account for slight differences in the image space between different sequences from the same patient. The final model is then trained on the full 4,817 MRI images together with the 50 CT images. As an ablation we also train a model only on the MRI images.

**Preprocessing** We used z-score normalization for all training steps. The images were resampled to the spacing given in Table 1

**Table 1.** Spacings used for resampling in the different trainings.

| Training | CT Descriptors | T1 images | All MRI | All MRI+CT |
|---|---|---|---|---|
| Spacing | $[2.5, 0.8, 0.8]$ | $[2.5, 0.75, 0.75]$ | $[2.6, 0.78, 0.78]$ | $[2.5, 0.78, 0.78]$ |

**Training:** We use the default configurations, generated by nnU-Net for all trainings. The respective patch sizes for each training are given in 2. All generated configurations consist of 6 resolution stages. We keep the batch size at 2 for all initial trainings to prevent overfitting on the small datasets and only increase it to 4 for the final trainings on the large dataset. Figure 1 shows a schematic overview of the generated network architeture.

**Table 2.** Patch sizes used for each training.

| Training | CT Descriptors | T1 images | All MRI | All MRI+CT |
|---|---|---|---|---|
| Spacing | 40x224x192 | 40x192x256 | 40x192x256 | 40x192x224 |

**Inference:** nnUNet's inference pipeline is not optimized for single image inference like it is the task in this challenge. We therefore make several small adjustments to the default pipeline to minimize resource usage and prediction time. First, we disable all test time augmentations and calculate the argmax directly on the raw logits instead of the softmax probabilities. Second, we swap the default *skimage*-based resampling function for the much faster *torch* resampling, significantly speeding up segmentation export in exchange for a slight loss in performance.

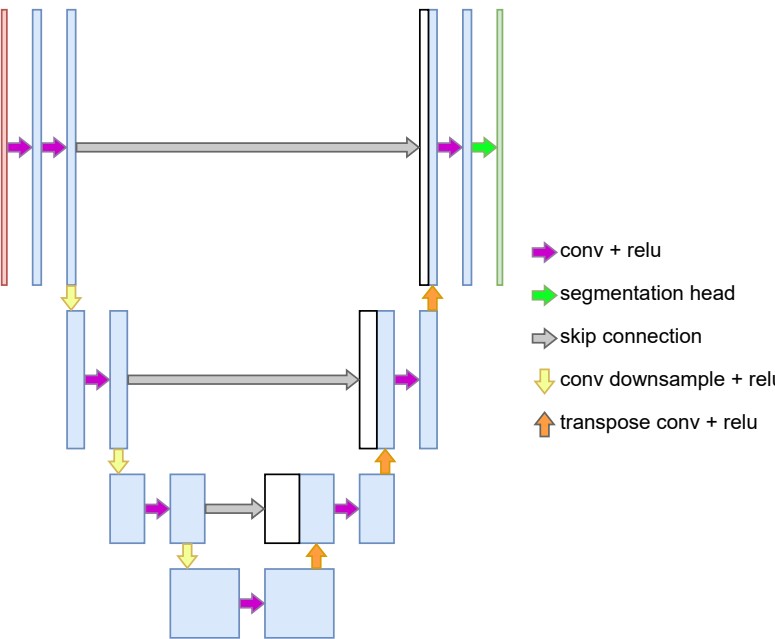

**Fig. 1.** Schematic network architecture of the U-Net created by nnU-Net's default configuration.

## 3   Experiments

### 3.1   Dataset and evaluation measures

The training dataset was curated from more than 30 medical centers under the license permission, including TCIA [2], LiTS [1], MSD [21], KiTS [?,9], autoPET [6,5], AMOS [13], LLD-MMRI [?], TotalSegmentator [25], and AbdomenCT-1K [20], and past FLARE Challenges [17,18,19]. The training set includes 2,050 abdomen CT scans and over 4,000 MRI scans. The validation and testing sets include 110 and 300 MRI scans, respectively, which cover various MRI sequences, such as T1, T2, DWI, and so on. The organ annotation process used ITK-SNAP [28], nnU-Net [11], MedSAM [15], and Slicer Plugins [4,16].

The evaluation metrics encompass two accuracy measures—Dice Similarity Coefficient (DSC) and Normalized Surface Dice (NSD)—alongside two efficiency measures—running time and area under the GPU memory-time curve. These metrics collectively contribute to the ranking computation. Furthermore, the running time and GPU memory consumption are considered within tolerances of 15 seconds and 4 GB, respectively.

### 3.2   Implementation details

**Environment settings** The development environments and requirements are presented in Table 3.

**Table 3.** Development environments and requirements.

| | |
|---|---|
| System | Ubuntu 20.04 |
| CPU | AMD Ryzen 9 3900X processor |
| RAM | 64GB DDR4-3600 RAM |
| GPU (number and type) | One NVIDIA RTX3090 GPU with 24GB VRAM |
| CUDA version | 12.1 |
| Programming language | Python 3.11 |
| Deep learning framework | torch 2.4.0 |

**Training protocols** We used the default nnU-Net pipeline of data augmentations, consisting of spatial - i.e. rotations, mirroring - and intensity transformations, without further modifications. The final models were selected by expected inference times and performance on the public validation set.

### 3.3   Test Set Submission

Task 3 of the FLARE challenge allowed for only one submissions to the final test set. We therefore submitted the model trained with isotropic spacing of 2.5mm, which showed better performance than the half resolution model on the public validation set (see table 5).

**Table 4.** Training protocols.

| | |
|---|---|
| Network initialization | random |
| Batch size | 4 |
| Patch size | 40×192×224 |
| Total epochs | 1000 |
| Optimizer | SGD |
| Initial learning rate (lr) | 1e-2 |
| Lr decay schedule | PolyLR Scheduler |
| Loss function | Soft Dice loss + Cross Entropy loss |
| Number of model parameters | 30.71M |

## 4    Results and Discussion

### 4.1    Quantitative results on validation set

The results of the final submission on the public validation set are shown in table 5. The model trained on the MRI images together with the 50 CT images generally performs better than the model trained only on the MRI images, with an increase of 3.1 points in DSC and 3.3 points in NSD. Only for the kidneys and aorta, training only on MRI images performs better than training on MRI and CT images together. For some classes, the MRI model seems to have significant problems in correctly segmenting the structures. This is especially apparent for the esophagus with a Dice of only 17.1, but also classes like the adrenal glands, pancreas, and duodenum seem to suffer from the modality transfer.

**Table 5.** Quantitative evaluation results of the submitted method trained on MRI and CT images and the ablation trained on MRI only on the public validation set.

| Target | Public Validation | | Public Validation (Ablation) | |
|---|---|---|---|---|
| | DSC(%) | NSD(%) | DSC(%) | NSD(%) |
| Liver | **87.2 ± 7.4** | **83.0 ± 13.1** | 86.5 ± 11.2 | 82.4 ± 15.2 |
| Right Kidney | 89.0 ± 9.4 | 84.6 ± 11.3 | **90.5 ± 8.9** | **87.0 ± 9.8** |
| Spleen | **63.2 ± 22.2** | **51.9 ± 23.8** | 52.4 ± 27.2 | 43.5 ± 26.0 |
| Pancreas | **37.6 ± 19.1** | **48.7 ± 23.0** | 35.7 ± 19.8 | 46.1 ± 24.1 |
| Aorta | 82.5 ± 13.4 | 84.3 ± 15.4 | **84.1 ± 12.4** | **85.8 ± 14.6** |
| Inferior vena cava | **47.3 ± 19.6** | **42.1 ± 19.5** | 43.5 ± 21.7 | 38.1 ± 20.9 |
| Right adrenal gland | **43.3 ± 17.7** | **59.8 ± 19.2** | 41.9 ± 20.5 | 57.3 ± 24.0 |
| Left adrenal gland | **35.9 ± 21.0** | **49.7 ± 22.3** | 30.6 ± 23.4 | 42.4 ± 27.4 |
| Gallbladder | **55.8 ± 27.7** | **43.3 ± 28.7** | 47.5 ± 30.5 | 35.9 ± 29.9 |
| Esophagus | **17.1 ± 17.4** | **28.6 ± 21.6** | 12.6 ± 16.2 | 21.9 ± 22.2 |
| Stomach | **60.6 ± 15.1** | **61.1 ± 15.6** | 55.8 ± 18.4 | 56.3 ± 19.0 |
| Duodenum | **39.1 ± 18.7** | **51.9 ± 21.3** | 36.2 ± 20.9 | 48.1 ± 24.3 |
| Left kidney | 91.9 ± 4.5 | 88.8 ± 6.0 | **92.7 ± 3.7** | **89.9 ± 6.4** |
| Average | **57.7 ± 9.2** | **59.8 ± 10.5** | 54.6 ± 10.7 | 56.5 ± 12.0 |

### 4.2 Qualitative results on validation set

Figure 2 shows qualitative results of the submitted methods on four cases from the public validation set. The submitted method generally performs well on most abdominal organs. However, the method tends to undersegment target structures. This is more pronounced in predictions from the model trained on MRI images only.

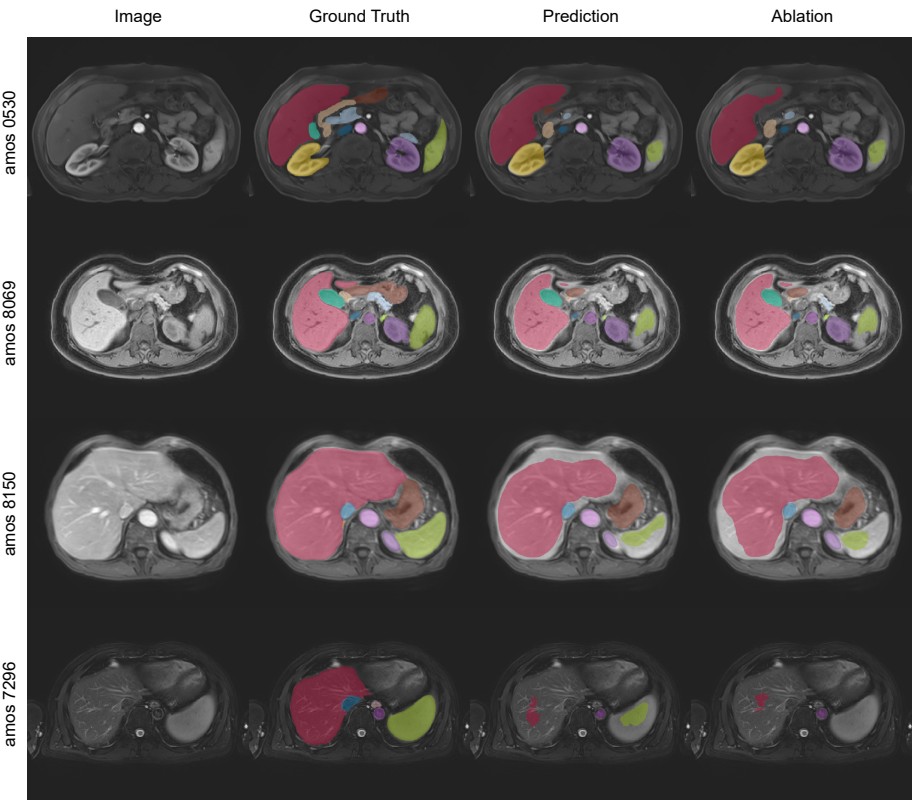

**Fig. 2.** Qualitative results of the submitted method, trained on MRI and CT images and the ablation trained on MRI only, on four example cases. The upper two rows show cases, where the model performs well, the lower two rows show examples of bad predictions and a near total failure, respectively.

### 4.3 Segmentation efficiency results on validation set

Table 6 shows running time and VRAM utilization of both submissions on 8 selected cases from the public validation set. The model complies with the time

limit for all of the 8 cases. However, as the testing was performed on a significantly better GPU, it is expected that the model might exceed the time limit for exceptionally large cases in the final testing.

**Table 6.** Quantitative evaluation of segmentation efficiency in terms of the running time and GPU memory consumption. Total GPU denotes the area under GPU Memory-Time curve. Evaluation GPU platform: NVIDIA RTX3090 (24G).

| Case ID | Image Size | Running Time (s) | Max GPU (MB) | Total GPU (MB) |
|---------|------------|------------------|--------------|----------------|
| amos_0507 | (320, 290, 72) | 15.6 | 5109 | 55128 |
| amos_0540 | (192, 192, 100) | 13.4 | 4980 | 44871 |
| amos_0546 | (576, 468, 72) | 19.9 | 5284 | 74444 |
| amos_0557 | (512, 152, 512) | 21.2 | 5204 | 75588 |
| amos_7236 | (400, 400, 115) | 19.7 | 5397 | 73898 |
| amos_7324 | (256, 256, 80) | 15.4 | 5087 | 54271 |
| amos_7799 | (432, 432, 40) | 23.6 | 5848 | 98605 |
| amos_8082 | (1024, 1024, 82) | 32.6 | 4927 | 112053 |

### 4.4   Results on final testing set

**Table 7.** Segmentation performance on the test set.

| DSC (%) | | NSD (%) | |
|---------|--------|---------|--------|
| Avgerage | Median | Avgerage | Median |
| $41.8 \pm 29.8$ | $56.6\,(7.7, 67.8)$ | $42.6 \pm 31.6$ | $56.5\,(3.7, 70.7)$ |

**Table 8.** Segmentation efficiency on the test set.

| Runtime (s) | | GPU (GB) | |
|-------------|--------|----------|--------|
| Avgerage | Median | Avgerage | Median |
| $19.1 \pm 4.9$ | $18.4\,(16.1, 20.6)$ | $1136.2 \pm 414.1$ | $1069.4\,(913.3, 1236.7)$ |

Tables 7 and 8 show the final results for segmentation performance and efficiency on the test set, respectively.

### 4.5   Limitation and future work

In our contribution, we relied on MIND descriptors to transfer the labels between modalities. These MIND descriptors should be modality independent, however,

they show differences, especially at the borders of structures. This might, for example, explain the observed undersegmentation of the final model. Including the CT scans for training at the earlier steps, especially when training on the T1 images only, might help with this issue, as inclusion of CT images seems to help with clearer boundaries.

An approach we briefly tried but did not pursue further is the registration of the CT images to the MRI images. We extracted the most similar CT-MRI pairs using perceptual hashing [26] and then applied the pretrained and further fine-tuned uniGradICON [23,22] on these pairs. The results looked very promising, however, in order to comply with the challenge rules, we could not use the pretrained model but instead had to train from scratch on the given CT and MRI data. Unfortunately, the results of these trainings were not convincing and we consequently dropped the idea.

## 5    Conclusion

In this paper, we addressed the challenge of abdominal organ segmentation on MRI scans, learning from labeled CT images, in the context of Task 3 of the FLARE 2024 challenge. Our approach to this task utilized nnU-Net, training multiple models to effectively transfer the labels from CT to MRI. The final model achieved competitive performance on the public validation set, however, it tends to undersegment and fails for some structures like the esophagus.

**Acknowledgements** The authors of this paper declare that the segmentation method they implemented for participation in the FLARE 2024 challenge has not used any pre-trained models nor additional datasets other than those provided by the organizers. The proposed solution is fully automatic without any manual intervention. We thank all data owners for making the CT scans publicly available and CodaLab [27] for hosting the challenge platform.
The present contribution is supported by the Helmholtz Association under the joint research school "HIDSS4Health – Helmholtz Information and Data Science School for Health". Part of this work was funded by Helmholtz Imaging (HI), a platform of the Helmholtz Incubator on Information and Data Science. This work was partially supported by RACOON, funded by "NUM 2.0" (FKZ: 01KX2121) as part of the RACOON Project.

## Disclosure of Interests

The authors declare no competing interests.

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
