# OpenReview forum: "Efficient Cross-Modality Abdominal Organ Segmentation Using nnU-Net and MIND Descriptors"
_MICCAI.org/2024/Challenge/FLARE — FLARE 2024 withMinorRevisions_

### Official Review · Reviewer_SNWN · 2025-01-17
**Review Comments**

**Rating:** 5
**Confidence:** 4

**Review:**

The authors propose a cross-modal abdominal organ segmentation method that combines nnU-Net and Modality-Independent Neighborhood Descriptor (MIND), demonstrating potential in multi-organ segmentation under resource constraints by transferring CT labels to MRI images. However, there are several notable weaknesses in the paper.

### 1. Lack of a Clear Framework Diagram：
The paper lacks a diagram that specifically presents the innovative method or main workflow of the proposed approach. Figure 1 only shows an abstract concept of nnU-Net and does not provide any details on the primary methods or processes introduced in the paper.

### 2. Limited Use of CT Labels：
The paper does not fully utilize CT labels, using only 50 labeled CT images. There is a lack of comparison with a baseline model trained using the full set of CT labels (2000+), which would help validate the effectiveness of the proposed method.

### 3. Insufficient Explanation of the Method：
The paper does not have a dedicated section explaining the innovative aspects of the proposed method. Instead, it only briefly discusses the training and prediction configurations without going into sufficient detail about the novelty and contributions of the approach.

These issues need to be addressed to improve the clarity and validity of the proposed method.

---

### Official Review · Reviewer_n7qr · 2025-01-24
**Comments**

**Rating:** 5
**Confidence:** 5

**Review:**

The authors introduce a novel framework for unsupervised segmentation that integrates two key components: (1) a segmentation model employing MIND descriptors to facilitate transfer learning, and (2) another segmentation model initially trained using label filtering and registration techniques. By integrating these advanced technologies, the method offers a solution for unsupervised domain adaptation (UDA) tasks.

However, several aspects of the framework require further elaboration:

1.	The overall pipeline diagram lacks clarity.

2.	Registration plays a crucial role in generating pseudo labels; what are the quantitative impacts of this process?

3.	Please provide detailed descriptions of the network architecture.

4.	The introduction does not adequately cover related works, and the description of the proposed method is unclear.

---

> ### Author Response · Authors · 2025-03-30
>
> Registration is only used when transferring the labels from the T1 images to the other sequences to account for small misalignments. Apart from that we experimented with registering the CT images to the MR images, but didn't pursue that further. The network architecture is described in quite some detail in the text and the schematic figure.

---

### Official Review · Reviewer_8uJu · 2025-02-16
**Using MIND descriptor for cross-modality segmentation**

**Rating:** 8
**Confidence:** 5

**Review:**

The proposed method leverages the nnU-Net framework combined with modality-independent neighborhood descriptor (MIND) features to transfer labels from CT to MRI images.  Overall, the method is technically sound and easy to follow.

---

### Author Response · Authors · 2025-03-30

We adapted the paper according to the reviewer comments.

---

### Decision · Program_Chairs · 2025-03-20

**Decision:**

Accept

**Comment:**

Please carefully address the reviewers' comments in the revision.